# Long-Term Dose Optimization of Adalimumab via Dose Spacing in Patients with Psoriasis

**DOI:** 10.3390/bioengineering9080387

**Published:** 2022-08-13

**Authors:** Michael Benzaquen, Mohammad Munshi, Simon Bossart, Laurence Feldmeyer, Vladimir Emelianov, Nikhil Yawalkar, Simone Cazzaniga, Kristine Heidemeyer

**Affiliations:** 1Department of Dermatology, Inselspital-Bern University Hospital, University of Bern, CH-3010 Bern, Switzerland; 2Centro Studi GISED, 24128 Bergamo, Italy

**Keywords:** inflammatory skin diseases, psoriasis, adalimumab, dose spacing, dose optimization, long term

## Abstract

Dose spacing (DS) can be useful for optimizing treatment with biologics in psoriasis patients. However, interval prolongation might increase the production of anti-drug antibodies (ADA) and, therefore, reduce the drug’s effectiveness. The long-term effects of DS with adalimumab in psoriatic patients have not been reported. The goal of our study was to evaluate the long-term follow-up of psoriatic patients after adalimumab DS regarding the clinical course and determination of circulating adalimumab, TNFα levels, and anti-adalimumab antibodies. We retrospectively included seven patients treated with adalimumab for moderate-to-severe psoriasis and benefiting from DS from 2010 to 2021. The dose interval of adalimumab was extended to three weeks for all patients and then to four weeks for three of the seven patients. Adalimumab trough levels, TNFα levels, and ADA against adalimumab were measured. For six of the seven patients, absolute PASI values remained below 3 throughout the follow-up period (median = 8.0 years; range: 1.7–11.5) after DS. All the patients were satisfied with the effectiveness of their treatment regime. Within the follow-up period, an average of 63 doses of adalimumab per patient were spared. The median adalimumab trough levels were 4.7 µg/mL (range: 1.9–12.5). TNFα levels remained under 10 pg/mL (undetectable) in all except one patient. ADA against adalimumab remained negative (<10 µg/mL) during the follow-up in all patients. Our data indicate that therapeutic drug monitoring, including the measurement of trough concentrations and ADA, together with the clinical response and patient’s preference, can be helpful for clinical decision making and treatment optimization in psoriasis.

## 1. Introduction

Dose spacing (DS), i.e., increasing the interval of drug administration, can be a useful option in treating psoriasis patients with biological therapies. The rationale for this off-label use of spaced doses is to find the minimal doses necessary to reach a good response while reducing potential side effects and relieving costs on health care systems [1,2]. Previous studies have proven the efficacy and safety of DS of biologics in treating psoriasis [1,3,4]. However, the long-term effect of DS with adalimumab in psoriatic patients over several years has not been reported. Therefore, the main objective of our study was to retrospectively evaluate the long-term follow-up of psoriatic patients after adalimumab DS and to correlate clinical status with circulating adalimumab, TNFα levels, and the presence of anti-adalimumab antibodies.

## 2. Methods

We retrospectively included seven patients treated with adalimumab for moderate-to-severe psoriasis and benefiting from DS from 2010 to 2021 at our Department of Dermatology, University Hospital in Bern, Switzerland. All adult patients with plaque psoriasis treated with an optimized dose of adalimumab who received at least one determination of adalimumab trough levels, TNFα levels, and ADA during the optimization period were included in the study. Three patients were excluded due to unclear intervals or shortening of intervals, and four patients were excluded due to lack of follow-up. Finally, 7 patients were considered for analysis. Adalimumab trough levels, TNFα levels, and ADA against adalimumab were measured at the Immunology and Allergy Laboratory of the University Hospital of Lausanne, Switzerland. The study was approved by the ethical committee of the Canton of Bern (KEK-2019-00313) and conducted according to the Declaration of Helsinki principles. All patients signed informed consent before inclusion in the study.

## 3. Results

The dose interval of adalimumab was extended to three weeks for all patients and then to four weeks for three of the seven patients. Patients’ demographics and clinical features are summarized in Table 1. The three-week DS was initiated after a median of 35 months of conventional adalimumab treatment every 2 weeks. At the time of DS, absolute PASI values were 0 or <1 for 70% of the patients and <3 for the remaining 30% of the patients. For six of the seven patients, absolute PASI values remained below 3 throughout the follow-up period (median = 8.0 years; range: 1.7–11.5) after DS. Only one patient showed an increase in skin psoriasis lesions and an absolute PASI value greater than 3 after 24 months. The median adalimumab trough levels were 4.7 µg/mL (range: 1.9–12.5) (Figure 1). Two patients had adalimumab trough levels ≤ 3 µg/mL. TNFα levels remained under 10 pg/mL, meaning undetectable, in all except one patient, for whom TNFα levels first reached 23.0 pg/mL before declining under 10 pg/mL after 5 months. Furthermore, ADA against adalimumab remained negative (<10 µg/mL) during the follow-up in all the patients during DS with adalimumab.

## 4. Discussion

Although we often use adalimumab in standard dosages for psoriasis patients, a dose-reduction strategy, particularly in patients with low disease activity and at their request, can prevent overtreatment without the risk of secondary loss of effectiveness over several years. The CONDOR study recently investigated a tightly controlled DS strategy in psoriasis patients with low, stable disease activity. Although noninferiority regarding disease activity was not demonstrated, DS of adalimumab, etanercept, and ustekinumab was possible in 53% of patients, without safety concerns [3]. Previous studies have also shown the safety of dose tapering other TNFα inhibitors in psoriasis [4,5]. Regarding infliximab, the data are contradictory, and DS cannot usually be recommended. The dose tapering of infliximab via interval prolongation led to a relapse of psoriasis in 25% of subjects in one study, while PASI 90 was maintained in all patients in another [6,7].

A recent study demonstrated that the disease-activity-guided dose adaption of TNFα inhibitors, including adalimumab, certolizumab pegol, etanercept, golimumab, and infliximab, is safe in the treatment of PsA, with no difference in disease activity [8]. Regarding the economic consequences, an analysis showed that this DS strategy resulted in a mean cost saving of EUR 3820 per patient over a period of 12 months with a minimal decrease in quality-adjusted life years [9].

However, interval prolongation might increase the production of anti-drug antibodies (ADA) and, therefore, jeopardize the drug’s effectiveness. Adalimumab is a fully humanized monoclonal IgG1 antibody and tumor necrosis factor α (TNFα) antagonist used in several indications, including moderate-to-severe plaque psoriasis and psoriatic arthritis. Previously, it has been shown that high titers of ADA to adalimumab were strongly correlated with undetectable adalimumab concentrations and with nonresponse or loss of response to adalimumab in patients with plaque psoriasis [10]. Moreover, it has been demonstrated that patients with no ADA formation in the first 24 weeks of treatment have a very low risk of developing these in the following 24 weeks. Dose interval shortening is also less useful in the presence of ADA [11]. Regarding the effectiveness of adalimumab, a therapeutic range of adalimumab trough levels of 3.51 µg/mL to 7.00 µg/mL corresponding to an optimal clinical effect has been defined [12], and these findings have further been confirmed by a multicenter study [13].

In our study, all the patients who underwent treatment with DS of adalimumab extending the drug interval to 3 or even 4 weeks were satisfied by the effect and applicability of the treatment of their psoriasis. Within the long-term follow-up period with a median of 8.0 years (range 1.7–11.5), six of the seven patients maintained stable, limited disease activity with PASI < 3. None of the three patients with PsA had arthralgia or further symptoms of PsA during the follow-up period. Within the follow-up period, an average of 63 doses of adalimumab per patient were spared in our seven patients (a total of 454 injections) by the end of this study. Adalimumab trough levels fell to less than or equal to 3 µg/mL in only two patients with a DS extending the drug interval to 4 weeks; however, these patients still experienced good efficacy of the DS regimen with adalimumab (absolute PASI < 3). TNFα is known to be higher in active psoriatic patients than in healthy patients [14], even though no standard values exist. In all of our patients, TNFα levels remained undetectable (<10 pg/mL), indicating a persistent control of systemic inflammation. In one patient, it only rose for a limited period (TNFα level 23.0 pg/mL) and declined spontaneously to an undetectable level without dose adaption within 5 months. During the whole follow-up period of 11 years, no patient developed ADA against adalimumab.

The detection of adalimumab trough levels and anti-adalimumab antibodies is a helpful method of monitoring the clinical recurrence of psoriasis after DS. Our data emphasize that the DS of adalimumab may not increase immunogenicity in psoriasis patients with low, stable disease activity over a long period of several years. Further studies are needed to better understand clinical and laboratory predictors for successful DS with adalimumab.

## Figures and Tables

**Figure 1 bioengineering-09-00387-f001:**
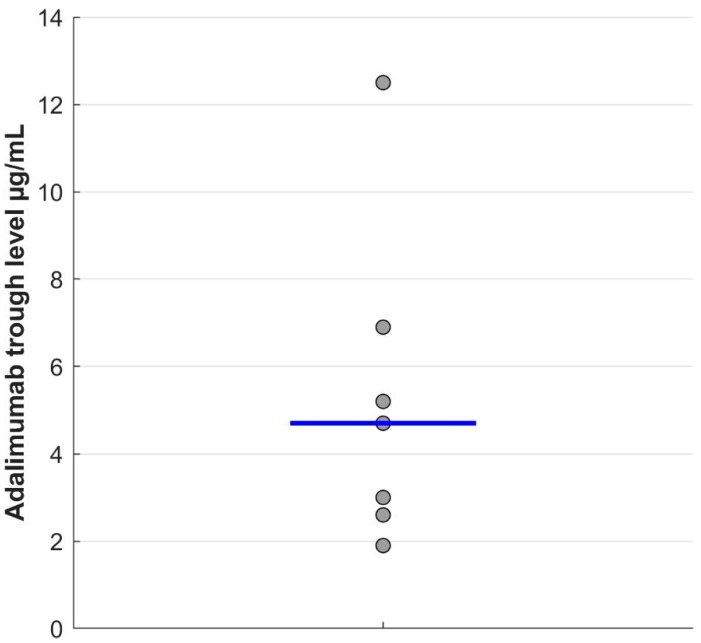
Adalimumab trough levels were measured after a median period of 18 months (range: 4–43) after starting DS. The blue line indicates the distribution median.

**Table 1 bioengineering-09-00387-t001:** Patient demographics.

		N (%)	Median (Range)
Sex	Male	6 (85.7%)	
	Female	1 (14.3%)	
Age (years)			61 (41–73)
Duration of Psoriasis Until Dose Spacing (Months)			35 (6–52)
BMI (kg/m^2^)			26.1 (22.2–40.1)
Previous Systemic Treatments *	Acitretin	6 (85.7%)	
	Methotrexate	6 (85.7%)	
	Efalizumab	2 (28.6%)	
	Alefacept	1 (14.3%)	
Presence of Psoriatic Arthritis **	Yes	3 (42.9%)	
	No	4 (57.1%)	
Use of Biosimilars **	Yes	2 (28.6%)	
	No	5 (71.4%)	

BMI: body mass index; * multiple treatments were possible; ** during study period.

## Data Availability

All the data generated or analyzed during this study are included in this article. Further inquiries can be directed to the corresponding author.

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
