# Peer review of "Long-Term Dose Optimization of Adalimumab via Dose Spacing in Patients with Psoriasis"

_bioengineering, 2022, doi:10.3390/bioengineering9080387_

Round 1

Reviewer 1 Report

1.TNFa < 10 pg/ml means undetectable level? That should be strictly defined.

2.What is the optimal level of serum TNF-a for justifying therapeutic effects of adalimumab? That should be defined.

3.Three patients have PSA. What about the activity of PSA during the DS study? Worsened or not?

4.Are there any previous studies about DS of other anti-TNF-a antibody, such as infliximab, etanercept, certolizumab pegol? Those should be cited and discussed.

Author Response

We thank the reviewer for his/her valuable comments. Please find our answers below.

1) TNFa < 10 pg/ml means undetectable level? That should be strictly defined.

Answer: TNFa < 10 pg/ml means undetectable level. This was confirmed by the laboratory (Service d’immunologie et d’allergologie -LIA, CHUV Centre Hospitalier Universitaire Vaudois, Suisse). We have added this explanation in the manuscript: “TNFα levels remained under 10 pg/mL, meaning undetectable” (line 70)

2) What is the optimal level of serum TNF-a for justifying therapeutic effects of adalimumab? That should be defined.

Answer: Contrary to Adalimumab level, there is no range of TNF-a level to be reached. TNF-a can also be measured in healthy non-psoriatic patients, but usually at a lower level. (Arican O, Aral M, Sasmaz S, Ciragil P. Serum levels of TNF-alpha, IFN-gamma, IL-6, IL-8, IL-12, IL-17, and IL-18 in patients with active psoriasis and correlation with disease severity. Mediators Inflamm. 2005 Oct 24;2005(5):273-9.). There are no standard values of serum TNF-a. However, a non-detectable serum TNF-a indicates a better control of the systemic inflammation.

We have added in the manuscript: “TNFα is known to be higher in active psoriatic patients compared to healthy patients13, even though no standard values exist. In all our patients TNFα-levels remained undetectable (<10 pg/mL), indicating a persistent control of systemic inflammation. In one patient it rose only for a limited period (TNFα-level 23.0 pg/ml) and declined spontaneously to an undetectable level without dose adaption within 5 months.” (lines 119-123)

3) Three patients have PSA. What about the activity of PSA during the DS study? Worsened or not?

Answer: None of the 3 patients with PsA developed arthralgia or other symptoms of PsA after dose reduction during the follow-up period. We have added this important fact in the manuscript: “None of the 3 patients with psoriatic arthritis (PsA) developed symptoms of active PsA during the follow-up.” (lines 67-68) and “None of the 3 patients with PsA had arthralgia or further symptoms of PsA during the follow-up period.” (lines 114-115)

4) Are there any previous studies about DS of other anti-TNF-a antibody, such as infliximab, etanercept, certolizumab pegol? Those should be cited and discussed.

Answer: We added a paragraph in the manuscript: “Previous studies have also shown the safety of dose tapering of other TNFα inhibitors in psoriasis.4 Regarding infliximab, the data are contradictory and DS cannot be usually recommended. Dose tapering of infliximab by interval prolongation leaded to a relapse of psoriasis in 25% of subjects in one study while PASI 90 was maintained in all patients in another. 5,6 A recent study demonstrated that disease activity-guided dose adaption of TNFα inhibitors including adalimumab, certolizumab pegol, etanercept, golimumab and infliximab are safe in the treatment of PsA with no difference in disease activity. 7” (lines 87-94)

Reviewer 2 Report

In this manuscript Benzaquen et al. retrospectively examined the clinical records, adalimumab trough levels and development of anti-drug antibodies in 7 patients who underwent dose spacing, in which the time interval for injection of adalimunumab was extended beyond the usual period to determine if such dose spacing increased the risk of developing anti-drug antibodies and/or resulted in poorer disease control. They found that most of the patients maintained disease control and did not develop anti-drug antibodies. Further, the adalimumab trough levels averaged about 4.7 µg/mL, thought to be within the effective range. Because such dose spacing results in cost savings and may lessen the possibility of adverse drug effects, the results suggest that dose spacing should be considered and possibly tested for psoriatic patients on adalimumab. The study is interesting but the primary weakness is the small number of patients examined. Indeed, even though only one patient seemed to lose disease control (with an increased PASI value above 3), with only 7 total patients, this winds up being ~15% of the population studied. As a pilot study, however, the results of this study suggest the possibility of impacting clinical care. A few minor suggestions for improving the manuscript are provided below.

Minor points:

(1) The authors should provide the TNFalpha levels measured in the patients.

(2) In line 33 "rational" should be "rationale". In line 47 "reveiced" should be "received". In line 100 "where" should be "were".

(3) The sentence in lines 105-17 should be edited. One possibility is: "Adalimumab trough levels fell to less than or equal to 3 µg/mL in only 2 patients with a DS to 4 weeks; however, these patients still experienced good efficacy of the DS regimen with adalimumab...."

Author Response

We thank the reviewer for his/her valuable comments. Please find our answers below.

1) The authors should provide the TNFalpha levels measured in the patients.

Answer: No TNFa- levels have been determined at baseline/ before treatment with adalimumab. During the dose spacing period, TNFa-levels remained <10 pg/ml, meaning undetectable according to the laboratory’s specification. Only in 1 patient this level has been measured at 23.0 pg/ml but was once again undetectable 5 months later. We have specified this in the manuscript: “TNFα-levels remained under 10 pg/mL, meaning undetectable, in all except one patient for whom TNFα-levels first reached 23.0 pg/ml before declining under 10 pg/ml after 5 months.” (lines 70-72).

2) In line 33 "rational" should be "rationale". In line 47 "reveiced" should be "received". In line 100 "where" should be "were".

Answer: We have corrected the manuscript accordingly.

Reviewer 3 Report

Nice article. In any case, I feel that it can be easily shortened to a research letter without losing the significant data, that is that 7 of the 14 patients with dose spacing were analized, and no antidrug antibodies were found probably due to the start of the optimization later than 24 months after the adalimumab start.

Llamas-Velasco and Daudén E (PMID: 32761730)in Dermatologic Therapy studied a larger group of patients  treated with adalimumab and your article results fits with their results. They did not test antidrug antibodies but they found that the control of the disease were good despite optimization that was started if patient were controlled in two consecutive visits. 

Biobadaderm study and Dra Baniandrés article, also brought similar results that must be included in your article as there are no so many published articles on dose optimization in psoriasis. 

Author Response

We thank the reviewer for his/her valuable comments. Please find our answers below.

1) Llamas-Velasco and Daudén E (PMID: 32761730) in Dermatologic Therapy studied a larger group of patients treated with adalimumab and your article results fits with their results. They did not test antidrug antibodies but they found that the control of the disease were good despite optimization that was started if patient were controlled in two consecutive visits.

Answer: We have added this reference in the manuscript: Ref. 5. Llamas-Velasco M, Daudén E. Reduced doses of biological therapies in psoriasis may increase efficiency without decreasing drug survival. Dermatol Ther. 2020 Nov;33(6):e14134

2) Biobadaderm study and Dra Baniandrés article also brought similar results that must be included in your article as there are no so many published articles on dose optimization in psoriasis.

Answer: We have not found any specific article regarding dose spacing in the Biobadaderm registry. We have already included the article from Dr Baniandrés in the manuscript: Ref. 7. Baniandrés O, Rodríguez-Soria VJ, Romero-Jiménez RM, Suárez R. Dose Modification in Biologic Therapy for Moderate to Severe Psoriasis: A Descriptive Analysis in a Clinical Practice Setting. Actas Dermosifiliogr. 2015 Sep;106(7):569-77.

Round 2

Reviewer 1 Report

The authors well addressed the issues I pointed out previously, and appropriately revised the manuscript.

Author Response

The authors well addressed the issues I pointed out previously, and appropriately revised manuscript

Answer: We thank the reviewer for his/her positive comment.

We also used English language editing service by MDPI.

Reviewer 3 Report

I think that it could be shortened without losing main message.

Author Response

We shortened the manuscript as much as possible without losing valuable information.